# Semen Quality Traits of Two Thai Native Chickens Producing a High and a Low of Semen Volumes

**DOI:** 10.3390/vetsci10020073

**Published:** 2023-01-18

**Authors:** Ngassa Julius Mussa, Wuttigrai Boonkum, Vibuntita Chankitisakul

**Affiliations:** 1Rukwa Region Commissioner’s Office, Sumbawanga 55180, Rukwa, Tanzania; 2Department of Animal Science, Faculty of Agriculture, Khon Kaen University, Khon Kaen 40002, Thailand; 3Network Center for Animal Breeding and Omics Research, Faculty of Agriculture, Khon Kaen University, Khon Kaen 40002, Thailand

**Keywords:** fresh semen, rooster, small farm holder, sperm quality

## Abstract

**Simple Summary:**

Smallholder farmers commonly encounter the problem of low fertility and hatchability after artificial insemination (AI). Because of that, they need to use the simple method as a tool to select potential cocks that could ensure high fertility in the breeding process. Our findings demonstrate that breeds are affected by sperm concentration. Thai roosters producing high semen volumes have superior sperm quality than those producing a low semen volume but not superior sperm viability. These findings suggest that the amount of semen production may be used as a tool in selecting Thai native roosters for breeding, especially for smallholder farmers who may not have access to specialized tools for semen evaluation.

**Abstract:**

Semen quality traits such as semen volume, sperm motility, sperm concentration, pH, and color are very important, since they can determine the fertility potential of the semen. The objective of this study was to compare the semen quality traits of Thai Native chickens (Pradu Hang Dam and Chee) producing high and low semen volume. Semen was collected from 24 roosters (12 reps) and divided into two groups of roosters producing high semen volume (>0.3 mL) and those producing a low amount of semen (<0.3 mL). Fresh semen quality parameters (semen volume, sperm motility, viability, concentration, and pH) were measured and compared between groups. It was found that semen volume showed a positive correlation with sperm concentration, mass movement, motility, viability, and pH of fresh semen. There was no significant difference in fresh semen parameters between Pradu Hang Dam and Chee roosters with either high or low ejaculate semen, except for sperm concentration. Sperm concentration was significantly higher in Chee compared with Pradu Hang Dam in both high and low ejaculate semen. The semen quality parameters between groups of chickens producing high and low ejaculate semen were significantly different except for sperm viability. In conclusion, it was found that chickens producing high volumes of semen have better sperm quality than those producing a low semen volume. Therefore, these results provide a possible consideration that the amount of semen production among chickens can be used as a tool in selecting Thai native roosters for breeding.

## 1. Introduction

As it has been reported in other places across the globe, chickens raised by farmers as smallholders frequently encounter the problem of low fertility and hatchability following artificial insemination (AI). Likewise, in Thailand, the inconsistency of the trend in the fertility percentage results among Thai native chickens following AI has been observed [1,2]. The association of sperm quality and fertility potential in the avian has been described and poor sperm quality reduces fertility and increases embryo mortality [3]. Therefore, the evaluation of semen quality parameters, such as volume, semen color, sperm concentration, sperm motility, and sperm viability, before AI, is very important in breeding roosters [4,5]. The relationship of these parameters provides excellent indicators of reproduction potential and usefulness in determining the fertility and hatchability of eggs [5,6]. Furthermore, semen volume, among other parameters assessed by simple or traditional methods, is widely used by smallholder farmers to select potential cocks that could ensure high fertility in the breeding process [7]. It was revealed that increases in the volume of ejaculate fluids give more space for sperm cells to move easily; thus, selecting roosters for higher ejaculate volume could also mean selecting them for higher sperm motility [8].

Despite their significant importance in infertility, reports from previous studies showed significant variations of experimental results in semen parameters, including the amount of ejaculate volume [9,10]. This phenomenon has resulted in an inconsistency in fertility potential following AI of either fresh or frozen semen [8,11]. Additionally, semen parameters, semen volume, and fertility in Thai native breeds have been observed to differ significantly [12]. However, studies on Thai native chicken semen report that factors such as breed and age do not influence these parameters [13]. Based on previous studies about the relationship between semen volume, motility, and fertility, semen volume facilitates sperm motility, and chickens producing high semen volume possess sperm with high motility because of the high volume of fluids provided [6,8,14]. Moreover, the amount of ejaculate volume among different breeds of native chickens differs significantly [9,15]. We hypothesize that semen volume can be a tool to indicate quality sperm among smallholder farmers. However, no report describes the influence of semen volume on sperm parameters and fertility in Thai native chickens. Therefore, we designed this study to compare the effect of semen production on the semen quality of Thai Native chickens producing high and low semen ejaculate volume.

## 2. Materials and Methods

### 2.1. Animals and Experimental Design

Thai native roosters from Pradu Hang Dam and Chee breeds (1.2–1.3 years of age, body weight at 3.71 ± 0.22 vs. 3.67 ± 0.27 kg, respectively) were managed intensively in a battery cage system, with 60 × 45 × 45 cm per rooster and 16 h light/day throughout the experiment. Each rooster received approximately 110 g feed/day, consisting of commercial breeder feed for male chickens (90.07% DM, 17.15% CP, 3.35% CF, 3.99% EE, and 9.75% Ash) and water ad libitum for 8 weeks.

The two breeds of a rooster were divided into 4 groups based on semen volume (Pradu High, Pradu Low, Chee High, Chee Low). However, semen volume in each rooster was evaluated on three occasions before dividing into two groups, producing high and low-volume ejaculates. Males with semen volume above or below 0.3 mL were designated as producing high and low-volume ejaculates, respectively. The designation of the given range was based on previous studies [13,16,17] that stated the average semen volume in Thai native chickens. A total of 24 roosters (*n* = 6 per group per breed) were included in the study. Experimental procedures were conducted after approval was given by the Institutional Animal Care and Use Committee based on the Ethics of Animal Experimentation of the National Research Council of Thailand (Approval No: 660201.2.11/95).

### 2.2. Semen Collection and Evaluation

Semen was collected twice a week for six (6) weeks consecutively via the dorsal abdomen massage method and diluted with 0.1 mL of the modified EK extender, as previously described [18]. Chickens were handled carefully during semen collection to prevent cross-contamination. The semen samples were protected from lights and kept at a temperature of 22–25 °C during transport to the laboratory within 20 min after collection for macroscopic and microscopic evaluation. For macroscopic evaluation, semen volume, color, and pH were determined, while for microscopic evaluation, sperm motility, viability, and sperm concentration were determined. All birds were classified into two groups based on either high ejaculate volume (≥0.3 mL) or low ejaculate volume (<0.3 mL) before being subjected to semen evaluation. All the procedures and assessments (sperm motility, viability, and concentration) were performed by the same experienced operators.

### 2.3. Semen Color, Volume, and pH

Semen volume was recorded directly on the microtube with 1.5 mL, while white and creamy color was selected for further analysis. Moreover, to determine the pH, pH indicator paper was immersed in a semen sample and removed quickly. After a few seconds, the color change was observed on the pH indicator paper, and the pH reading was recorded.

### 2.4. Sperm Motility

Sperm motility was classified as a mass movement and progressive motility. To determine the mass movement, one drop of semen sample was placed on a slide without a coverslip, examined under a compound microscope (100×) and scored into 1–5 scales. To determine the progressive motility, 5 µL of semen sample diluted with 100 µL of 0.9% sodium chloride was estimated by microscopic observation at 400× magnification. A total of 200 sperm were counted in at least 5 microscopic fields to obtain the final reading. The progressive motility was presented in terms of motile sperm percentage.

### 2.5. Sperm Viability

Sperm viability was assessed using the eosin-nigrosin staining method. Briefly, a 10 µL drop of fresh semen sample was placed on the slide, followed by a drop of 20 µL eosin-nigrosin, then gently mixed. This stain was left to dry (air dryer) for some minutes. At least 200 sperm were counted under a compound microscope (×400 magnification) to determine the percentage of live sperm. The stained sperm were considered dead sperm, while non-stained sperm were considered to live ones, and the results were expressed in terms of percentages.

### 2.6. Sperm Concentration

The sperm concentration was determined using a hemocytometer chamber. Five µL of semen sample was diluted with 195 µL of sodium chloride. A drop of semen sample was put on a hemocytometer, and the reading was recorded under a compound microscope (×400 magnification). Sperm concentration was expressed as million (10^6^) sperm cells/mL.

### 2.7. Statistical Analysis

The study was conducted at the experimental farm of Khon Kean University using a 2 × 2 factorial experiment in a Completely Randomized Design with two factors and twelve replications. The first factor was treatment, which included high and low ejaculate semen. The second factor was the Thai native chicken rooster breeds, which were two breeds: Pradu Hang Dam and Chee chicken roosters. Therefore, for each semen parameter, there were a total of 72 observation values for use in the statistical analysis. Duncan’s New Multiple Range Test was used to test the differences in semen quality parameters, and *p* < 0.05 was considered statistically significant. Twelve replications were used for evaluating semen quality. The results were analyzed using the statistical software program SAS 9.0. The full statistical model was as follows:$$Y_{ijk}=\mu +Breed_{i}+Volume_{j}+Breed_{i}\times Volume_{j}+\varepsilon_{ijk}$$
where

$Y_{ijk}$ = observation values from treatment combination at breed *i* (*i* = 1 to 2) and volume *j* (*j* = 1 to 2), and replication *k* (*k* = 1 to 12)μ = overall mean$Breed_{i}$ = the main factor of rooster breeds *i* (*i* = 1 to 2)$Volume_{j}$ = the main factor of semen volume *j* (*j* = 1 to 2)$Breed_{i}\times Volume_{j}$ = the interaction between rooster breeds and semen volume *ij* (*i* = 1 to 2 and *j* = 1 to 2)$\varepsilon_{ijk}$ = experimental error

## 3. Results

A summary of Pearson’s correlation (r) coefficient of semen volume, sperm concentration, mass movement, progressive motility, viability, and pH of fresh semen in two Thai native chicken breeds is shown in Table 1. The correlation of semen volume with all sperm quality was significantly positive (*p* < 0.05). Sperm motility in terms of mass movement (r = 0.303) and progressive motility (r = 0.484) was moderately positive. Moreover, the pH of semen showed a positive correlation with semen volume (r = 0.417) and progressive motility (r = 0.407).

The interaction between breeds and semen volume on each fresh semen parameter is presented in Table 2. The results demonstrated that rooster breeds (Pradu Hang Dam and Chee) and semen volume (low and high) independently influenced all semen parameters in terms of sperm concentration, mass movement, progressive motility, viability, and pH (*p* > 0.05). Each main effect was therefore interpreted separately. For the effect of breed on semen quality, the results indicated that only the sperm concentration was significantly higher in Chee compared with Pradu Hang Dam (3,347 vs. 3,074 × 10^6^ sperm cells/mL; *p* < 0.01), while the semen volume (0.29 vs. 0.28 mL), mass movement (3.98 vs. 3.91), sperm motility (81.26 vs. 81.10%), and pH (6.98 vs. 6.97) in Pradu Hang Dam and Chee did not show a significant difference (*p* > 0.05).

The semen quality parameters between groups of chickens producing high and low ejaculate semen were significantly different except for sperm viability (Table 2). Semen volumes (0.35 vs. 0.22 mL), mass movement (4.16 vs. 3.73), progressive motility (86.83 vs. 75.53%), sperm concentration (3291 vs. 3129 × 10^6^ sperm cells/mL), and pH (7.05 vs. 6.90) were significantly (*p* < 0.05) higher in chickens producing high ejaculate semen than in chickens producing lower ejaculate semen; meanwhile, sperm viability was not different (83.09 vs. 83.00%; *p* > 0.05).

The comparison between Pradu Hang Dam and Chee roosters with high and low ejaculate semen was demonstrated in Figure 1. There was no significant difference in fresh semen parameters in terms of semen volume, mass movement, progressive motility, sperm viability, and pH between Pradu Hang Dam and Chee roosters with either high or low ejaculate semen (Figure 1A,C–F), except sperm concentration (Figure 1B). Chee had higher (*p* < 0.05) sperm concentration compared with Pradu Hang Dam in both high and low ejaculate semen.

## 4. Discussion

We successfully attempted to create populations of chickens producing high and low ejaculate volumes. This is the first study comparing semen parameters between chickens producing a high semen volume with those with low semen in Thai native chicken breeds. There were no differences in semen quality between breeds except in sperm concentration. Most semen quality parameters (except sperm viability) in a group of chickens producing a higher amount of semen volume than those with a low amount were significantly higher. In contrast, sperm viability did not differ significantly. The relationship between semen volume and other sperm parameters, such as sperm motility, viability sperm concentration, semen color, and pH, determines successful fertility.

Semen quality is important in facilitating AI and semen cryopreservation. As in other species, assessing semen quality in poultry before being used for AI is required because of its significant effects on AI and fertility [10]. Moreover, semen quality and quantity are affected by breeds and strains of chickens [19,20]. In the present study, the effects of breeds on semen quality, as well as the comparison of semen quality between Pradu Hang Dam and Chee roosters with high and low ejaculate semen, show that there were no differences in sperm motility, viability, pH, and volume between breeds except for sperm concentration, which was higher in Chee than in Pradu Hang Dam. Several studies on the semen characteristics of indigenous chickens have been conducted. Tarif et al. [10] evaluated semen characteristics of four indigenous breed lines in Bangladesh and found a non-significant difference in other semen characteristics, except for sperm concentration. Peters et al. [8] observed differences in strains in the semen concentration of Nigerian indigenous chickens. Moreover, the mean value concentrations obtained in the present study are consistent with those reported in previous studies of indigenous chickens [6,21]. Therefore, it was described that differences in sperm concentration among different chicken breeds and lines might be attributed to several factors, such as breed differences, genetic variation, individual performance, and stimulation [22]. On the other hand, similarities of the other semen parameters in the present study have been previously studied and associated with factors such as genetic and management factors. Usman et al. [23] studied genetic diversity in five Iranian native chicken populations and observed differences in functional genes among chicken breeds. It was suggested that the differences in functional genes were due to differences in genetic origins, breeding systems, sample sizes, and chicken types. Additionally, when semen of four Korean native chickens of similar genetic origin and White Leghorn (Commercial breed) were determined, morphological defects in Korean native chickens were higher than in the White Leghorn [3]. Therefore, we speculate that similarities in semen quality parameters observed between Pradu Hang Dam and Chee breeds in the present study can probably be due to similar genetic originals, as these chickens are also genetically related and domesticated under the same breeding system [24].

In the present study, we observed higher mass movement, progressive motility, and concentration in a group of chickens producing a high amount of semen volume than those producing a low amount. Studies of semen characteristics among different native chicken breeds have been reported. Moreover, their results have been observed to vary significantly among chickens [25,26]. The relationship between semen volume and other sperm parameters, such as concentration and motility, has been established. It has been revealed that the increase in semen volume increases sperm fluidity, increasing sperm movement [8]. This relationship has been confirmed in our results between the chicken groups, as we observed higher semen quality in chickens producing higher semen volume than those producing low semen volume. Additionally, several studies suggested that the relationship between semen production and sperm quality and fertility in chickens exists. Churchil et al. [27] and Das et al. [28] found a positive correlation between the semen volume of Rhode Island Red roosters and live sperm. Additionally, Parker [29] suggested that the decline in semen production in chickens during the summer and fall months is evidence that semen production is associated with semen quality and fertility in chickens. Conversely, the fecundity rate was reported to be high in chickens producing a high volume [30]. This evidence strengthens our hypothesis about the influence of semen production in Thai native chickens.

Moreover, semen pH was significantly different between groups of chickens producing high and low ejaculate semen. Chickens producing semen with a high volume had a higher semen pH than those producing low semen volume. It is well-accepted that semen pH correlates with sperm quality characteristics such as motility, viability, concentration, and volume; for instance, an acidic environment can directly affect sperm quality [31]. In goat and human sperm, decreased pH was observed to reduce sperm concentration, motility, and viability [31,32]. Our results found that the pH in chickens producing a high volume is slightly alkaline (7.05). In contrast, in those producing a low amount of semen, the volume is slightly acidic (6.90). The effects of pH on sperm quality in birds have been investigated. The pH recorded by previous researchers, particularly in roosters, also showed a positive correlation with sperm quality [26,33]. Therefore, in the present study, pH in chickens producing a high ejaculate volume is within these ranges. The reduced semen quality in a group of chickens producing a low amount of semen volume in our study was because of lower pH.

However, the sperm viability and semen color between groups of chickens producing high and low ejaculate semen did not differ significantly. Previous authors have reported that in chickens, the proportion of viable sperm varies from 83.3–89.0%, 80.0–96.0%, and 82.2–87.3% [4,10,22], respectively. It is reported that the variations in the proportion of live sperm might be due to genetic variation in tolerance to diluents or disturbances during transport to the laboratory [10]. In the present study, this scenario probably does not apply. Nevertheless, the proportion of viable sperm in the present study in both groups was good enough for routine AI in poultry. Semen color in the present study was creamy-white, indicating that the massage technique used in the study may be acceptable for a chicken semen collection to obtain good quality semen for AI. Creamy-white semen in this study was consistent with [8,33], who reported creamy-white semen in purebred naked neck Tswana and black Nigerian indigenous naked neck cockerels, respectively. It is reported that variations in semen color may arise partly due to contaminants, such as urine, feces, blood, or low sperm concentration [15].

According to [26], chickens with heavy body weight produce high amounts of semen. However, in our study, the body weight of Pradu and Chee was not different (3.71 ± 0.22 vs. 3.67 ± 0.27 kg). It might be due to the similarity of those stains (heavy breeds; matured weight > 2.5 kg), and they were raised under the same environmental factors. The selection of chickens in both groups of high and low semen volume, therefore, could not consider using this criterion. The semen quality might be evaluated individually.

## 5. Conclusions

In conclusion, since this is the first study to compare the effect of semen production and sperm quality of different Thai native chickens, the present study found that chickens producing a high amount of semen have better sperm quality than those producing a low semen volume. Therefore, these results provide a possibility considering that the amount of semen production among chickens can be used to select roosters for breeding. Considering the effect of age on semen production, the proposed age of roosters in this study may be suggested to farmers.

## Figures and Tables

**Figure 1 vetsci-10-00073-f001:**
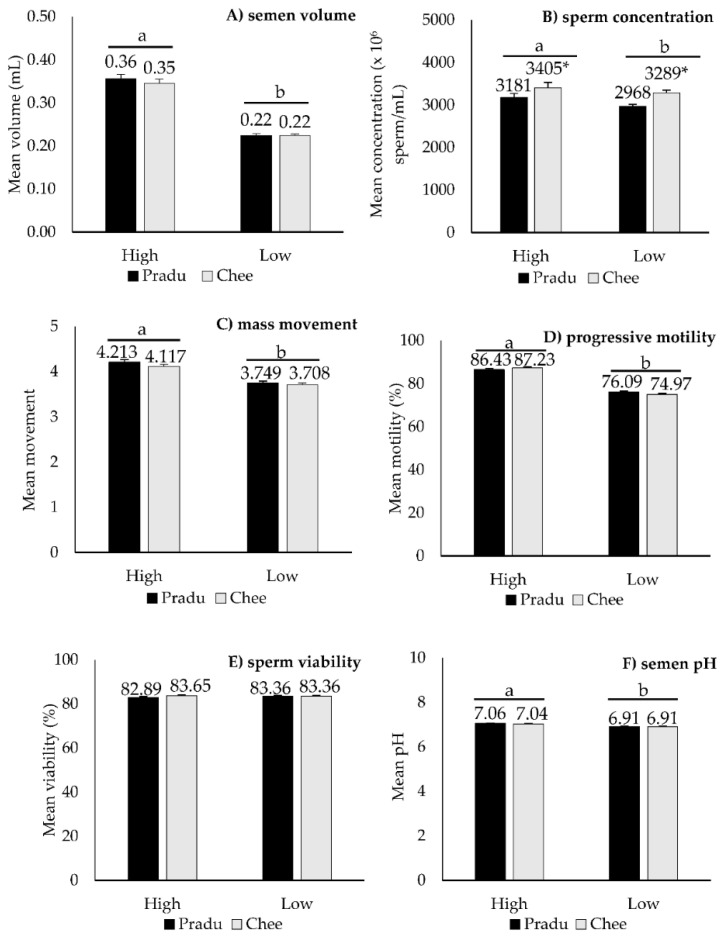
Comparison between Pradu Hang Dam and Chee roosters with high and low ejaculate semen (mean ± SE) on (**A**) semen volume, (**B**) sperm concentration, (**C**) mass movement, (**D**) progressive motility, (**E**) sperm viability, and (**F**) semen pH. Means between the high and low groups with superscript letters a and b indicate significant differences (*p* < 0.05). Means between chicken breeds in each high and low group with the superscript symbol * indicate significant differences (*p* < 0.05).

**Table 1 vetsci-10-00073-t001:** Pearson’s correlation coefficient (above number) of semen volume, sperm concentration, mass movement, progressive motility, sperm viability, and semen pH of fresh semen in two Thai native chicken breeds and other semen parameters and *p*-value (below number).

Parameters	Semen Volume	Sperm Concentration	Mass Movement	Progressive Motility	Sperm Viability	Semen pH
Semen volume	-	0.110 (0.031)	0.303 (0.001)	0.484 (0.001)	0.124 (0.015)	0.417 (0.001)
Sperm concentration		-	0.189 (0.001)	0.254 (0.001)	−0.078 (0.126)	−0.061 (0.228)
Mass movement			-	0.604 (0.001)	−0.003 (0.949)	0.285 (0.001)
Progressive motility				-	0.030 (0.558)	0.407 (0.001)
Sperm viability					-	−0.043 (0.398)
Semen pH						-

**Table 2 vetsci-10-00073-t002:** Interaction effects between breeds and semen volume (*p*-values).

Parameters	Semen Volume	Sperm Concentration	Mass Movement	Progressive Motility	Sperm Viability	Semen pH
Breeds ^a^	ns	**	ns	ns	ns	ns
Volume ^b^	**	*	**	**	ns	**
Interaction	ns	ns	ns	ns	ns	ns

^a^ Pradu Hang Dam and Chee roosters. ^b^ high ejaculate volume (≥0.3 mL) and low ejaculate volume (<0.3 mL). * = *p* < 0.05; ** = *p* < 0.01; ns = non-significant (*p* > 0.05).

## Data Availability

The data presented in this study are available on request from the Network Center for Animal Breeding and Omics Research, Faculty of Agriculture, Khon Kaen University, Thailand.

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
