# Peer review of "Semen Quality Traits of Two Thai Native Chickens Producing a High and a Low of Semen Volumes"

_vetsci, 2023, doi:10.3390/vetsci10020073_

Round 1

Reviewer 1 Report

The manuscript provides new data on semen production and quality in Thai native chicken of scientific interest to add knowledge on the phenotypic characterization of semen in rare chicken breeds. However, the aim of the study is different and focused on results not innovative, being already known the positive relation among sperm quality parameters in chicken semen.

I suggest a different presentation of the data to underline their novelty and provide a mechanism to have high fertility after artificial insemination of fresh semen. Data recorded could be used for phenotypic characterization of semen production and quality in the Thai native chicken and the variability within and between breeds could be studied (descriptive statics, analysis of variance, correlation coefficients). The expected results will be useful to identify the potential threshold in semen volume required to reach the best fertility after in vivo artificial insemination, knowing the required standard dose per hen is 100 millions fresh sperm (Saver B, 1988. Reproduction des volailles et production d'oeufs, INRA. pp 209-228).

I not recommend the publication of the manuscript in the present form.

Reviewer 2 Report

Major comments: 

Thai native roosters from Pradu Hang Dam and Chee breeds of 1.2-1.3 years of age are considered in this study. 

Please comment on the conclusions from this study related to the age of the roosters. 

In section 2.7,  Duncan’s New Multiple Range Test was used to test the differences in semen quality parameters, and p<0.05 was considered statistically significant. A total of 24 roosters (n = 6 per group per breed) were included in the study. The standard deviation may change with the number of samples.  Please comment on the present study's conclusions about the number of samples or roosters considered.  Include the standard deviation for Chee roosters in Fig. 1a. 

Minor comments: 

In section 3, please describe the values in bold in the Table 1 caption. 

On page 7, include the observations for two different pH values considered in this study. 

Reviewer 3 Report

Manuscript ID: vetsci-2056434

To The Editors and the Authors.

In this original paper, the Authors found that semen volume, in Thai roosters, is positively correlated with fresh semen quality parameters.

In my opinion, I agree with the Authors that a simple tool, such as the assessment of sperm volume, could be helpful in small farms, to select good breeders, if a positive correlation with sperm quality parameters is assessed. So, I think that this manuscript could provide an improvement in the field of roosters breeding management.

However, I have a few notes regarding the article presented.

Altogether, I suggest an accurate English revision by a native speaker.

Lines 11 -18 – simple summary. This section should be addressed to a lay audience, but in my opinion, it is not clear and does not provide a simple and exhaustive summary of ms. I suggest rewriting it.

Line 38 – Introduction. This section is well-structured and provides an adequate description of the current state of the field. Also, aims and scopes are clearly stated. However, I suggest moving the last sentence (lines 70-71) to the discussion and conclusion section or, alternatively, rewriting it, as, to me, it seems that your hypothesis has already been demonstrated.

Line 72 – Materials and methods. The section is well structured and overall, procedures are clearly described. I only have some concerns about the objectivity of the performed analysis. Most of the assessments are performed by operators (sperm motility, viability, and concentration); a computerized analysis (CASA Systems, photometers) could have provided an objective result, while the described techniques provided subjective results from the operator’s opinion. Knowing that computerized tools are not always available, I suggest specifying if all procedures were performed by the same (or a group of) experienced operators.

More, in “animals and experiment design” (lines 73-88), is not clearly stated that animals were divided into groups in 4 groups basing on breed and semen volume (Pradu High, Pradu Low, Chee High, Chee Low)

Line 134 – results. I suggest adding also raw data of sperm analysis obtained from each group, at least in form of a table containing ranges and mean±DS obtained from the assessment of motility, viability etc…

The discussion and conclusions are well-written and provide complete information.
